# Wearable Technology and Visual Reality Application for Healthcare Systems

Kuang-Hao Lin * and Bo-Xun Peng

Department of Electrical Engineering, National Formosa University, Yunlin 632, Taiwan; 11065131@gm.nfu.edu.tw
* Correspondence: khlin@nfu.edu.tw

**Abstract:** This study developed a virtual reality interactive game with smart wireless wearable technology for healthcare of elderly users. The proposed wearable system uses its intelligent and wireless features to collect electromyography signals and upload them to a cloud database for further analysis. The electromyography signals are then analyzed for the users' muscle fatigue, health, strength, and other physiological conditions. The average slope maximum So and Chan (ASM S & C) algorithm is integrated in the proposed system to effectively detect the quantity of electromyography peaks, and the accuracy is as high as 95%. The proposed system can promote the health conditions of elderly users, and motivate them to acquire new knowledge of science and technology.

**Keywords:** wearable; IoT; VR; EMG; healthcare





## 1. Introduction

With the advances of modern medicine, people's average lifespan has prolonged; hence, the aging of population structures has become a global issue. Moreover, the changes of lifestyles have induced various chronic diseases, increasing the demand for medical care. As routine health information can enhance disease prevention and medical care, the application of telecare in routine health care can promote people's quality of life. In telecare, the first-aid devices can also collect personal physiological information for ambulances or relevant medical units, so that the patients under care can receive instant and effective assistance.

Literatures [1,2] used electrocardiography (ECG) combined with the Internet of Things (IoT) to design a remote monitoring and telemetry system that can improve the accuracy and reliability of heart health monitoring. Literatures [3,4] used electroencephalography (EEG) to design a sleep monitoring system and an epilepsy prediction system. Bhowmick used photoplethysmography (PPG) to design a PPG monitoring system and stored the PPG model parameters in the cloud for clinical diagnosis [5]. Huang used electromyography (EMG) to design a diet monitoring system, which receives EMGs of the mastication muscles through glasses to detect intake-related events [6].

This study combined physiological signal modules (namely, ECG, EEG, PPG, and EMG) with innovative educational technologies, such as the Internet, IoT [7,8], and virtual reality (VR) [9,10], to develop a cloud system with telecare services [11,12]. Using the VR and open-source software (OSS), this study developed an interesting VR digital game that could stimulate the elderly users to utilize the application for health management [13–15]. The usage scenario of the proposed VR health game is shown in Figure 1.

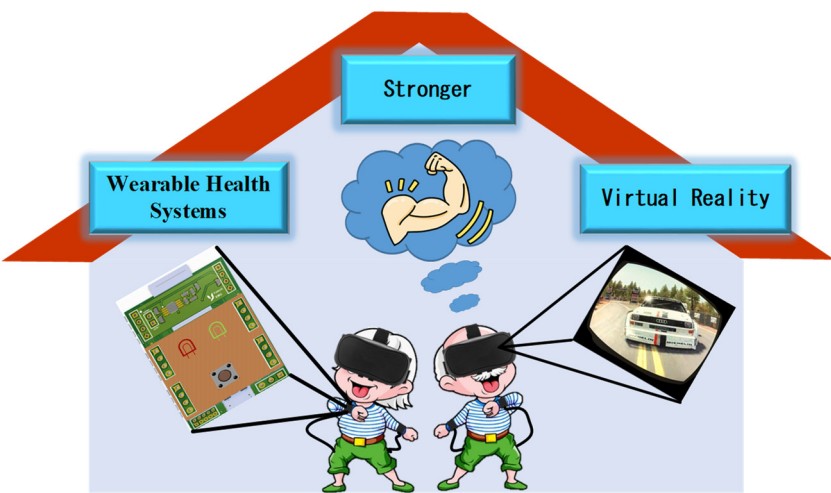

**Figure 1.** Schematic diagram of a system usage scenario.

Artificial intelligence (AI) has been widely used in the identification and analysis of physiological signals in recent years. Recent common EMG analyses and studies focused on AI application. For example, Sugiarto [15] analyzed the surface EMG (sEMG) and high-density EMG (HD-EMG) of the neck muscle (designated muscle) and used Convolutional Neural Network (CNN) for training to correct the end-to-end delay of the VR system and alleviate negative effects such as motion sickness. Sugiarto [15] also employed three pairs of wireless sEMG sensors from Delsys Trigno (Delsys, MA, USA) as the software. Pancholi [16] analyzed the peak average power (PAP) of EMG and adopted Linear Discriminant Analysis (LDA) and Quadratic Discriminant Analysis (QDA) for training and prosthetics. The analysis of Pancholi [16] has been done using MATLAB 2015a on an i7 core. Raurale [17] employed subsequently classified to classify and identify eight kinds of EMG actions. The system is also shown to operate in real-time on an ARM Cortex A-53 embedded processor suitable for housing in an EMG wearable device. Additionally, Wang [18] modified the So and Chan algorithm to increase the accuracy of ECG peak detection to 99.16% from 94.61% and used FPGA for verification. However, it is complicated and not applicable to biomedical VR games, which are characterized by low complexity [19,20]. Therefore, this study aims to develop an innovative realization architecture with low complexity. The proposed system specifically targets the irregularities of EMG peaks, in order to improve the operational accuracy and smoothness of VR games. To verify the EMG peak detection accuracy, this study first used the automatic calibration detection (ACD) to capture the EMC signals, and set a threshold parameter to assess the captured signals. To further improve the EMG peak detection accuracy, this study applied the average slope maximum So and Chan (ASM S & C) algorithm [21–25]. The comparison of recent studies on physiological signal detection systems is as shown in Table 1. The operation sensitivity was thus enhanced in the proposed VR game.

**Table 1.** Comparison of Recent Studies on Physiological Signal Detection Systems.

| Refrence | Algorithm | Complexity | Equipment |
|----------|-----------|------------|-----------|
| Sugiarto [15] | CNN | High | Delsys Trigno IMU |
| Pancholi [16] | LDA | High | I7 core |
| Raurale [17] | Subsequently classified | Medium | ARM Cortex A-53 |
| Wang [18] | Enhanced So and Chan | Low | FPGA |
| Our propose | ASM S&C | Low | nRF52840 |

The remainder of the paper is organized as follows. Section 2 describes the system architecture design and its specifications. Section 3 discusses the detailed design methodology for EMG peak detection. Section 4 presents the analysis of the experimental results. Finally, conclusions are drawn in Section 5.

## 2. System Architecture

*Hardware Architecture*

Figure 2 shows the hardware architecture of the smart wearable biomedical sensing system developed in this study. The sensing end uses the Tri-BLE, an IoT development platform, as the main body, and is provided with a Tri-EMG sensor, a button, and an indicator light. The human EMG signal is sent via a disposable electrode to the Tri-EMG sensor, and subsequently exported. The Tri-BLE platform filters the received EMG data, and the threshold of the EMG signal is judged according to the relationship between the EMG signal and the threshold. The output state is transferred to the central controller and processed. Furthermore, the button controls a calibration mechanism. The threshold of the EMG signal varies with the human body and the electrode position, and this threshold can be adjusted according to the indicated action after the button is pressed. The indicator light displays the state of the sensing end—the light is red during calibration, and the light turns green if the EMG signal exceeds the threshold. The central controller integrates the states of the left- and right-hand sensing ends, and communicates with the VR device to control the action in the game. Additionally, the system has a mobile app that is linked via Bluetooth. The muscle conditions of the left and right hands can be recorded, and accessed by doctors for future diagnoses.

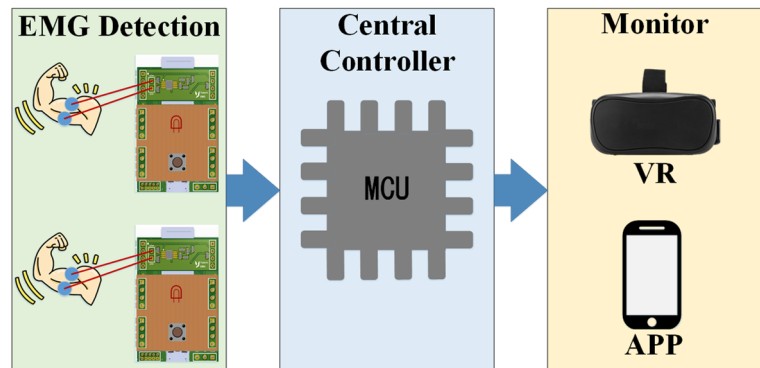

**Figure 2.** Hardware architecture of the smart wearable biomedical sensing system.

Table 2 lists the equipment specifications. Tri-BLE and Tri-EMG of TriAnswer series are used for the biomedical development platform, Arduino Uno is used as the central controller, and the monitor set includes an HTC VIVE VR headset and an app, with the associated equipment of HTC-VIVE and Android smartphone, respectively.

**Table 2.** Equipment specifications.

| System Requirement | Equipment |
| --- | --- |
| Biomedical Development Platform | TriAnswer (Tri-BLE, Tri-EMG) |
| Central controller | Arduino UNO |
| Monitor | Virtual reality HTC-VIVE Cosmos Android Smartphones |

## 3. EMG Peak Detection

### 3.1. System Flow

Figure 3 presents the moving average system flowchart. Notably, the human EMG signals are complicated and susceptible to external influences. This study thus employed the moving average and ACD methods to reduce the number of signal judgment errors. For the moving average, the EMG signals are imported at 500 Hz. The length of each data is 8 bits, and the system output is the average of 20 data points. Therefore, the first in first out (FIFO) buffer which can store 20 data points was used. When the buffer was filled with 20 data points, the average value was calculated as the output $Y(n)$.

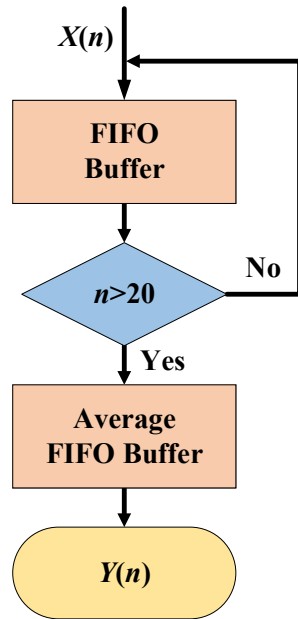

**Figure 3.** Moving average system flowchart.

Figure 4 is the ACD system flowchart. The ACD system adopts 10,000 EMG signals as the threshold calibration criteria. First, the input data are compared with the eight data in the buffer. When the input is larger than the buffer, the minimum value in the buffer is replaced by the new input data. After the system compares the 10,000 data points, the three largest data points are removed from the buffer to avoid the signal errors resulting in a high threshold. After five data points are averaged, this value is subtracted by the initial value ($O_{ini}$) and multiplied by the scale factor of $D$ to obtain the threshold after calibration.

Figure 5 is the average slope maximum So and Chan (ASM S & C) system flowchart. The *initial_value* is calculated first, and if the EMG signal exceeds the *initial_value*, the *initial_slope_maxi* is being searched. The *slope_threshold* is then obtained by the *initial_slope_maxi*. If two consecutive EMG slopes exceeds the *slope_threshold*, the current signal is recorded as the starting point of the waveform characteristics, and the peak could be determined. Finally, the peak and starting points are used for updating the *maxi*. The ASM S & C algorithm is described in Section 3.2.

### 3.2. Algorithm

EMG signal shows the potential difference when a muscle contracts. According to the state of motion, the amplitude and frequency of the electromyogram change accordingly. Under ideal conditions, an electromyogram indicates no movement, while obvious fluctuations occur when the muscles are contracted (as shown in Figure 6). However, EMG signals are susceptible to random interference caused by radio waves, electrode noise, and other factors [26–28].

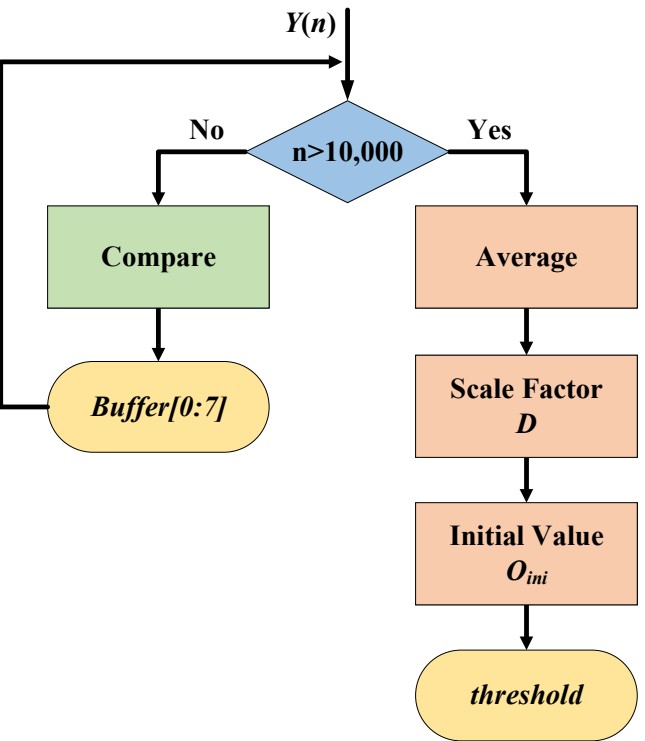

**Figure 4.** ACD system flowchart.

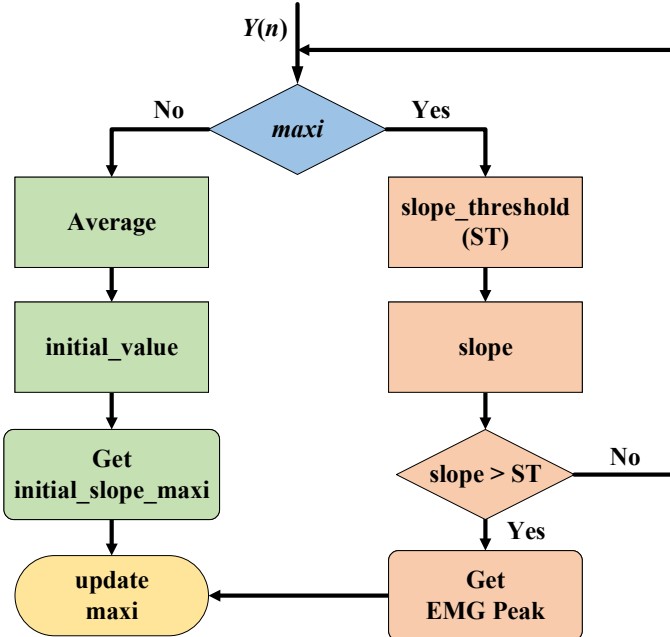

**Figure 5.** ASM S & C system flowchart.

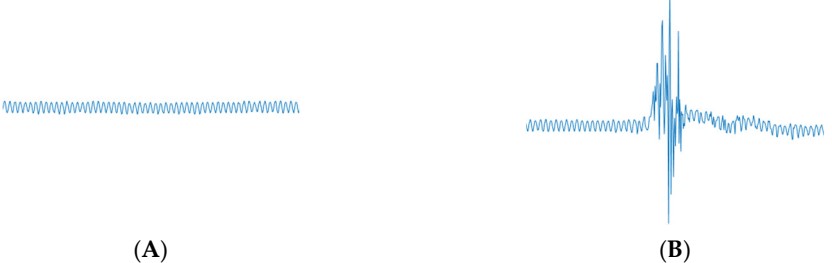

**Figure 6.** EMG comparison with and without movement. (**A**) No action. (**B**) Action.

The two considerations of noise interference and hardware design allow the moving average algorithm to effectively eliminate random interference in a large amount of data. The moving average algorithm used in this paper adopted feedback control as the filtering method. After a fixed period size is given, the output is the average of the input values in the period size. Its operating principle is to remove the oldest element and add the next input value before averaging the output. This output value divides the random noise evenly and makes it smaller, thus making the overall data waveform smoother.

The moving average algorithm is expressed in the following equation:

$$Y = \frac{\left[\sum_{P=0}^{P=T-1} X(n-P)\right]}{T} \tag{1}$$

where $Y$ is the output of the moving average; $T$ is the period length; and $X(n)$ is the current input data.

Although the moving average can effectively remove random noise, EMG changes can be easily read and compared with the threshold signal. However, factors such as different users or patch positions can change the EMG amplitude and cause a large gap in the threshold. Therefore, this paper designed an automatic calibration detection (ACD) algorithm to solve the problem of threshold error. The ACD algorithm collects 10,000 pieces of data to set the threshold. The ACD algorithm equation is as follows:

$$threshold = \left[\frac{1}{k}\sum_{n=1}^{n=k} S_{(n+3)}\right] \times D + [(1-D) \times O_{ini}] \tag{2}$$

where $S$ is the output after comparison, $k = m - 3$, $D$ is the scale factor, and $O_{ini}$ is the initial EMG value. A method similar to the sorting algorithm is used in the comparison. As long as the input value of the ACD algorithm is greater than the value in the buffer, it can be directly replaced. This method can find the largest $m$ data and sort them from the largest to the smallest. In order to prevent noise influence, the largest three pieces of data are removed from the $m$ data. There are signal bounces occasionally in the process of EMG signal sensing; hence, the top three of the eight pieces of data are removed to prevent the values of high signal bounces from affecting the average value.

The data averaged is multiplied by the scale factor $D$ to improve the accuracy of discrimination. Finally, $O_{ini}$ is added as the initial compensation of the threshold. The scale factor $D$ can obtain the best parameters after the numerical statistical analysis, as shown in Figure 7. When the scale factor is 50%, the accuracy of the EMG peak detection can reach 71%.

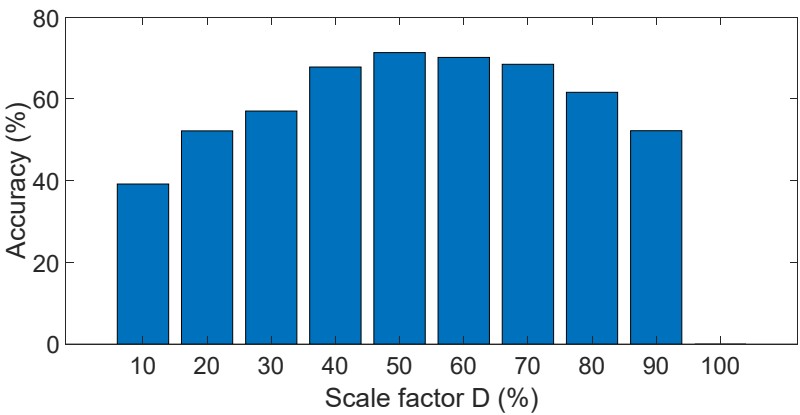

**Figure 7.** Scale factor statistical analysis.

The optimal scale factor in the ACD algorithm varies with the users; therefore, the data from each individual user should be statistically analyzed, and the scale factor cannot be corrected during real-time hardware variations. As a result, the accuracy varies by each user. In order to enhance the accuracy in detection, this study detected the EMG peaks, and used the transient peak when the muscle put forth strength as the reference frame for the overall threshold. This study adopted the So and Chan (S & C) algorithm, which is the R wave detection algorithm proposed by So and Chan in 1997 [22], and modified it into the average slope maximum So and Chan (ASM S & C) algorithm.

The S & C algorithm is given below [22]:

*Y(n)* is the processed EMG amplitude, and the slope of each point can be found in the time domain by Equation (3):

$$slope(n) = -2Y(n-2) - Y(n-1) + Y(n+1) + 2Y(n+2) \tag{3}$$

The *slope_threshold* can be obtained by Equation (4):

$$slope\_threshold = \frac{threshold\_paprameter}{16} \times maxi \tag{4}$$

First, a sample quantity is set up according to the sample rate of signal reception to find the *initial_slope_maxi*. The sample rate used in this paper is 500 Hz, as shown in Algorithm 1: S & C algorithm.

---

**Algorithm 1:** S & C Algorithm.

---

**Input:**
    *Y(n)*
**Output:**
  1: slope = 0
  2: *initial_slope_maxi* = 0
  3: for i: = 1 to 500 do
  4:      if i ≥ 3 then
  5:          slope = −2 × Y(i − 2) − Y(i − 1) + Y(i + 1) + 2 × Y(i + 2)
  6:         if slope > *initial_slope_maxi* then
  7:              *initial_slope_maxi* = slope
  8:         end if
  9:      end if
  10: end for
  11: maxi = *initial_slope_maxi*

---

The *slope_threshold* is determined by using the initial maximum slope. When two consecutive slopes are larger than the *slope_threshold*, this point is marked as the starting

point of the signature waveform, and then the peak point of waveform could be determined. The maxi is then updated, as expressed in Equation (5):

$$maxi = \frac{first\_maxi - maxi}{filiter\_parameter} + maxi \tag{5}$$

$$first\_maxi = \text{Height of p\_point} - \text{Height of start\_point} \tag{6}$$

For the *filiter_parameter* which can be set as 2, 4, 8, or 16, this paper selected 16. The peak was recorded, and the peak detection accuracy of the algorithm was calculated [22]. There were 100,000 EMG samples, the number of making a fist was 154, SP was the number of correct peaks captured successfully, MP was the number of correct peaks that were missed, and EP was the position of capture error. The accuracy can be computed by Equation (7). The S & C algorithm captured 154 peaks successfully, but there were 54 capture errors, for an accuracy of 74%.

$$\text{Accuracy}(\%) = \frac{\text{SP}}{\text{SP} + \text{MP} + \text{EP}} \times 100 \tag{7}$$

Figure 8 simulates the peak detection of the S & C algorithm. It was observed that the S & C algorithm had no obvious waveform characteristics in the first 500 samples, indicating there were errors in capturing the *initial_slope_maxi* and leading to errors in the subsequent *slope_threshold* and peak detection.

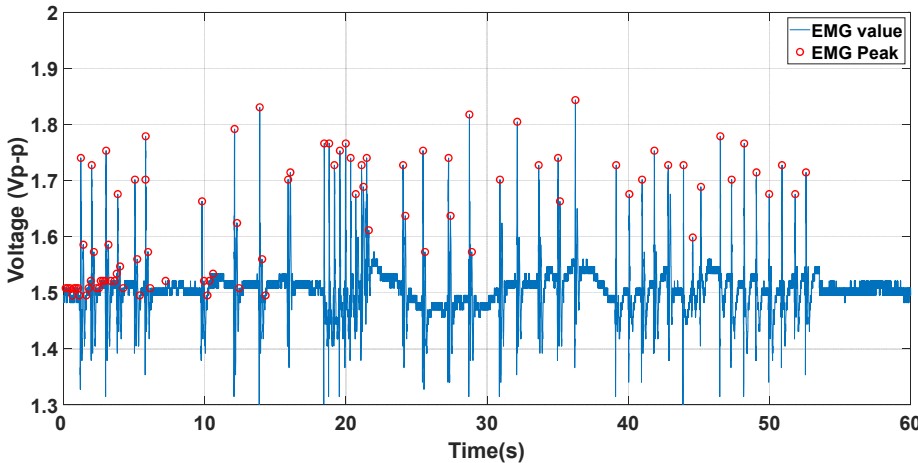

**Figure 8.** Peak detection of the S & C algorithm.

The S & C algorithm is mainly used for ECG detection. The main difference between EMG and ECG signals is that an EMG signal has obvious waveform characteristics only if the muscle contracts, whereas an ECG signal always has waveform characteristics. Hence, the *initial_slope_maxi* capture requirements of the S & C algorithm are not applicable to EMG. The *initial_slope_maxi* in the S & C algorithm is captured in a preset period of cycle time. In ECG with a cycle that changes slightly, accurate *initial_slope_maxi* can be obtained from the S & C algorithm after adjusting the cycle based on the ECG signals of different persons. However, for EMG signals with irregular changes, detection based on a fixed cycle is not feasible. In order to solve this problem, the capture requirements of the maximum slope were changed in this paper, as shown in Algorithm 2: ASM S & C algorithm.

**Algorithm 2:** ASM S & C Algorithm.

**Input:**
    *Y(n)*
**Output:**
  1: *initial_value* = avg(sum(Y(1:n))) × (λ + 1)
  2: slope = 0
  3: *initial_slope_maxi* = 0
  4: count = 0
  5: en = 0
  6: for i:= 1 to length(Y) do
  7:        if I ≥ 3 && Y(i) > *initial_value* then
  8:            *initial_slope_maxi* = −2 × Y(I — 2)−Y(I — 1)+Y(I + 1)+2 × Y(I + 2)
  9:            count = count + i
  10:          break
  11:        end if
  12: end for
  13: for i:= count to length(Y) do
  14:        slope = −2 × Y(I — 2) — Y(I — 1) + Y(I + 1) + 2 × Y(I + 2)
  15:        if slope > *initial_slope_maxi* then
  16:            en = 1
  17:        end if
  18:        if en == 1 then
  19:            if slope < *initial_slope_maxi* then
  20:               *initial_slope_maxi* = old_slope
  21:               break
  22:            end if
  23:        end if
  24: end for
  25: maxi = *initial_slope_maxi*

As the initial value of the EMG signal was not zero, the initial maximum slope capture requirements were changed in this study by using the average of *n* data and multiplying it by reduction ratio λ as the *initial_value*. The calculation approach could be expressed as Equation (8) in which *n* is set as 1000 samples. Each sample was 3.3 ms, and 1000 pieces of data (about 3.3 s) were captured for analysis during the initial setting of the *initial_value*, so as to capture stable *initial_value* data. When the detected signal exceeds this *initial_value*, it means that muscles have contracted and there are waveform characteristics. After determining the *initial_slope_maxi*, the subsequent *slope_threshold* calculation and peak detection can be performed.

$$initial\_value = \frac{(\lambda + 1)}{n} \times \sum_{1}^{n} Y(n) \tag{8}$$

λ in the *initial_value* calculation is the gathered statistics parameter. In order to avoid the error signal exceeding the average initial value, the *initial_value* calculation was multiplied by λ + 1 to evade the error signal. As only the error signal needed to be prevented, the accuracy was enhanced and stabilized after increasing the *initial_value* by 1%. An excessive λ value could increase the *initial_value* and result in the failure to capture the peak and a reduction of accuracy. Λ was thus selected as 5% of the stable zone after the statistical analysis, as shown in Figure 9.

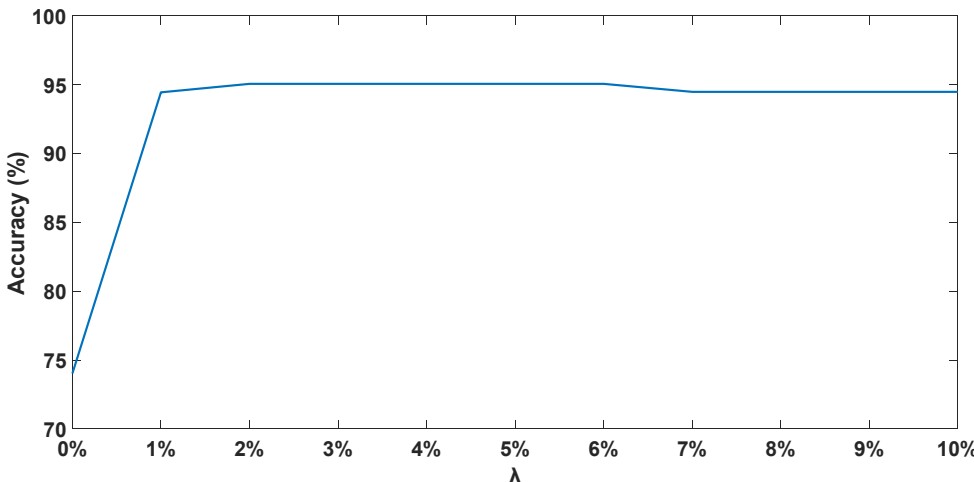

**Figure 9.** λ statistical analysis.

Figure 10 illustrates a simulation of the peak detection by the ASM S & C algorithm, which employs the *initial_value* calculation to solve the initial detection error in this study. The *initial_slope_maxi* was captured when the detected signal exceeded the EMG initial value. The ASM S & C captured 154 peaks successfully and there were eight capture errors, leading to an accuracy of 95.06%.

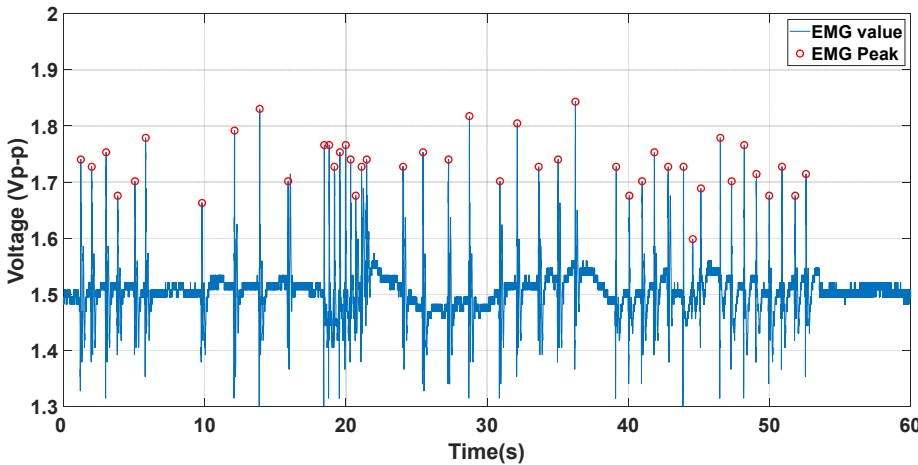

**Figure 10.** Peak detection of the ASM S & C algorithm.

Table 3 compares the accuracy of the three algorithms. Equation (7) was used for the calculation of accuracy. While the S & C and ASM S & C algorithms could capture all of the correct peaks, incorrect values were captured as well. The ASM S & C algorithm improved the accuracy of the S & C algorithm in capturing the EMG's initial maximum slope, reduced the subsequent error rate, and resulted in a final accuracy of 95.06%. Therefore, the accuracy of the ASM S & C method proposed in this paper was higher than both the S & C and ACD methods, by 20.98% and 23.71%, respectively.

**Table 3.** Algorithm accuracy comparison.

| Algorithm | SP (Data) | EP (Data) | MP (Data) | Accuracy (%) |
|-----------|-----------|-----------|-----------|--------------|
| ACD | 132 | 31 | 22 | 71.35 |
| S & C [22] | 154 | 54 | 0 | 74.08 |
| ASM S & C | 154 | 9 | 0 | 95.06 |

## 4. Experimental Results

### 4.1. Electromyography

In muscle contractions, the signal generated by the potential difference between both ends is called the EMG signal. Regarding the non-invasive measurement method used in this paper, disposable electrodes were attached to the skin to observe muscle activities. The experiment data were the muscle states of making a fist once per second within one minute, and the experiment platform was the designed sensing end of this system. The disposable electrodes were attached to the flexor digitorum superficialis muscle (FDS) and the flexor digitorum profundus muscle (FDP), as shown in Figure 11.

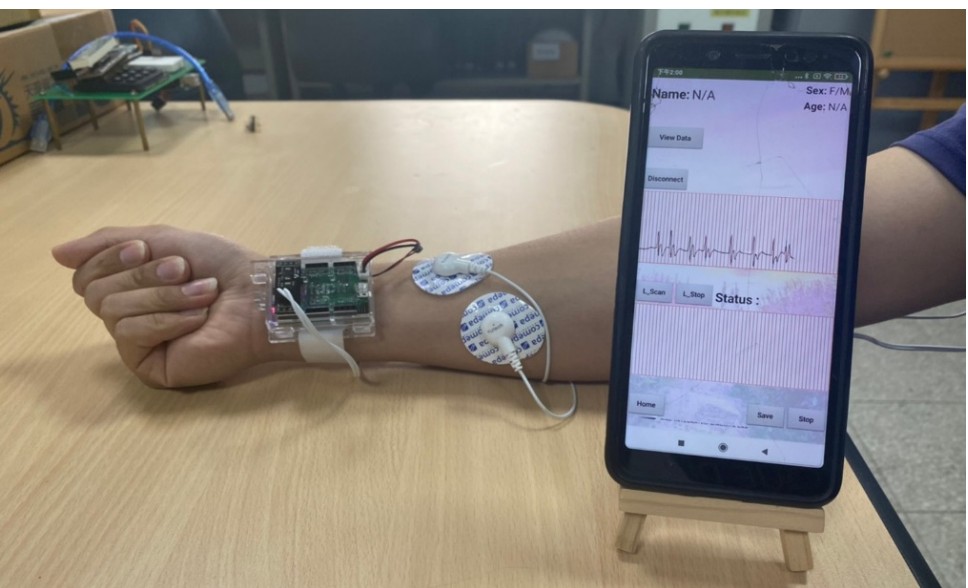

**Figure 11.** Electrode position stereogram.

The TriAnswer platform was strapped to the user's wrist to capture EMG signals via a patch. The captured EMG signals were transmitted to the cloud via the Bluetooth interface and displayed in a smartphone app.

The original signal and the filtered signal are shown in Figure 12. Apparently, the curve was smoothened after filtering. In Figure 12, EMG signals are obviously cleaner after filtration, which could improve the detection ability of the ASM S & C algorithm.

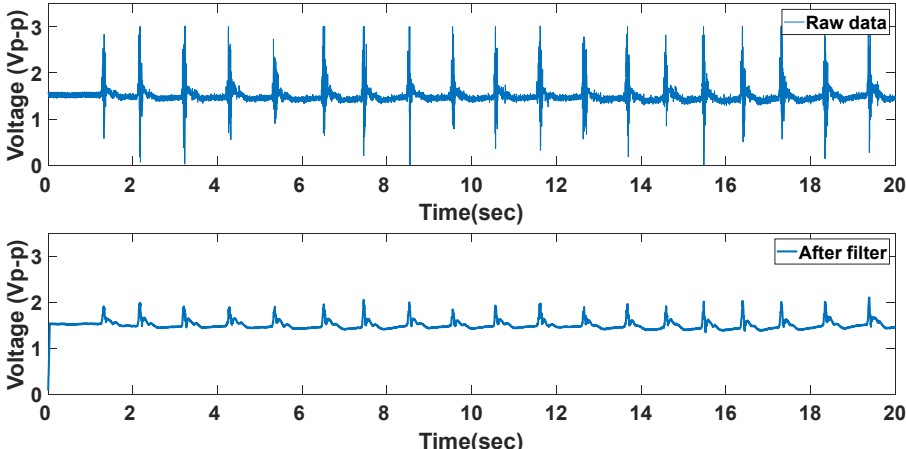

**Figure 12.** EMG filtering cross reference.

Figure 13 shows the spectrum analysis of Figure 12. Figure 13a presents the spectrum distribution of the EMG signals captured. Figure 13b displays the spectrum distribution of

the EMG signals after moving average. As seen, there are obviously fewer noises in the spectrum distribution below 60 Hz after the moving average.

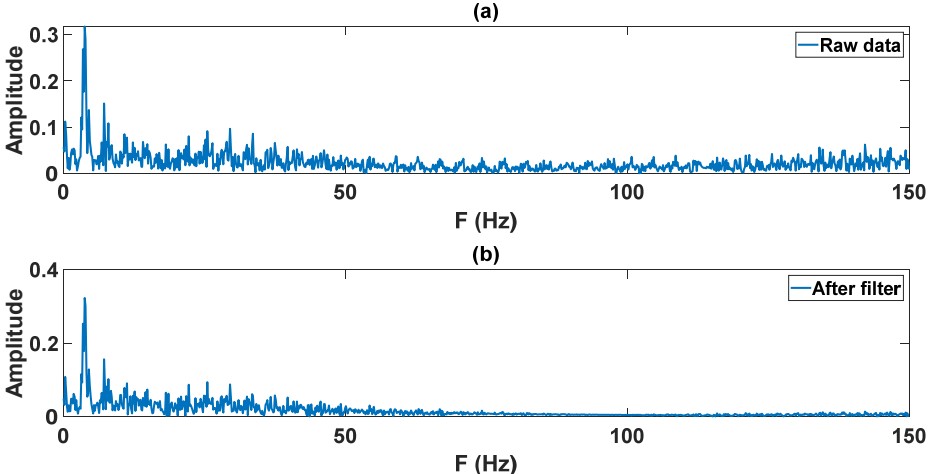

**Figure 13.** Spectrum analysis of the EMG signals. (**a**) presents the spectrum distribution of the EMG signals captured. (**b**) displays the spectrum distribution of the EMG signals after moving average

The proposed system controls the actions of the VR games by making a fist. When an EMG peak is detected, the output is 1; otherwise, the peak detection is 0. The experiment data are the same as the data for the moving average. The ASM S & C algorithm is used to determine the *initial_value* first, which is then used to find the *initial_slope_maxi*, as well as the *slope_threshold*. Whenever two consecutive EMG slopes exceed the *slope_threshold*, the peak can be identified and the maxi can be updated. The peak pulse is generated based on the peak, and the VR game is controlled by the peak pulse. Figure 14 shows the ASM S & C algorithm detecting the EMG signal and generating the peak pulse. The blue line is the EMG value, the pink dotted line is the initial value of the EMG, and the red dotted line is the peak pulse generated after the peak is detected. The peak pulse is used for controlling the actions of the VR figure.

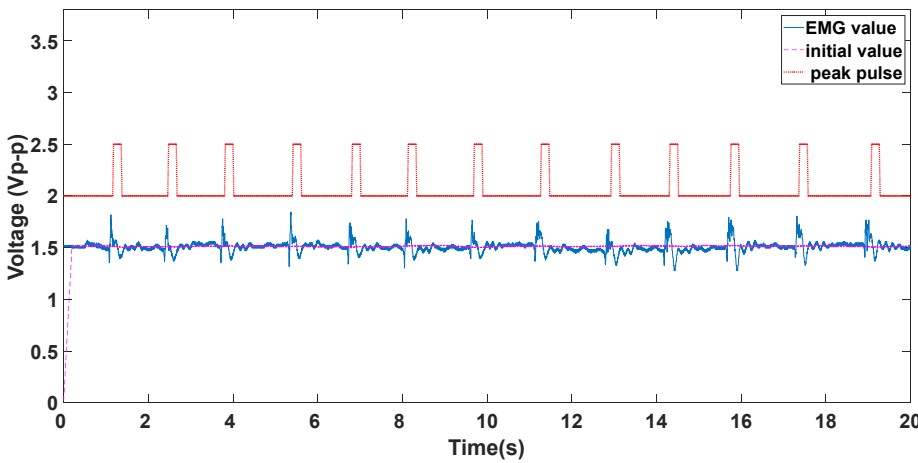

**Figure 14.** Peak pulse generated with the EMG signal.

### 4.2. Virtual Reality

VR can make the users feel like they are at the scene, making it feasible for training and fitness uses. In this study, VR was combined with health care to make health management more interesting. This system, which employs an HTC VIVE VR headset, differs from other systems that it uses EMG for game control. As shown in Figure 15, the serial port needs to be selected to connect with the control terminal before the game could start. When the

game starts, the direction is controlled by the left and right hands to evade obstacles, and the speed increases as the game progresses. The game ends when the player collides with an obstacle. The front-viewing angle displays the current time and the best time. The game picture is shown in Figure 16.

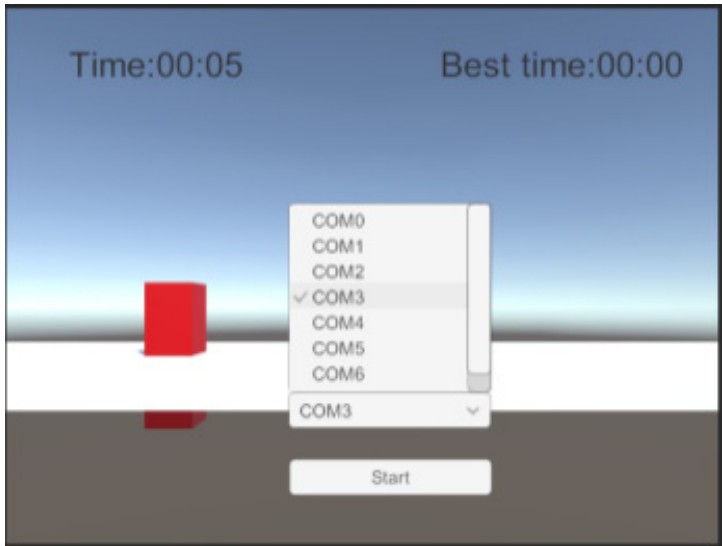

**Figure 15.** Game setting.

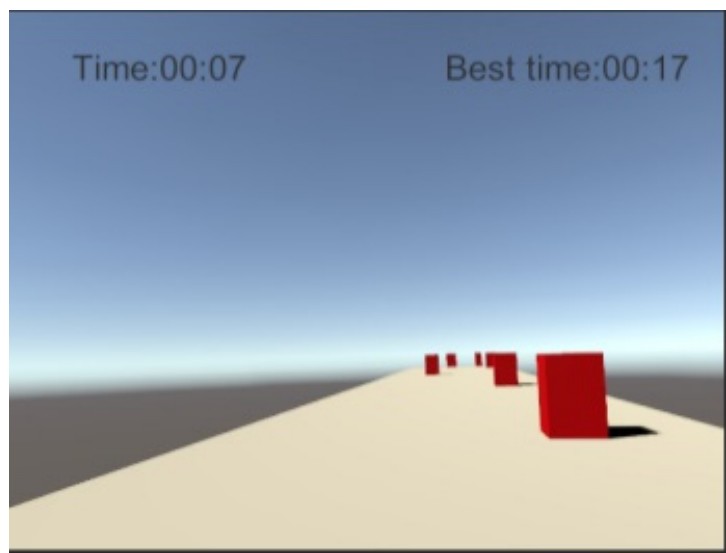

**Figure 16.** Game picture.

Figure 17 is the VR system flowchart. The Bluetooth COM port needs to be connected at the beginning of the game. When the game starts running, the EMG state is detected. When the left-hand EMG signal trigger is detected, the game figure would walk to the left; when the right-hand EMG signal trigger is detected, the figure would dodge to the right. The screen is updated periodically. When the game figure encounters an obstacle, whether the playtime exceeds the record is determined; if yes, the best time would be updated and displayed; if not, the best time would remain and the game would end.

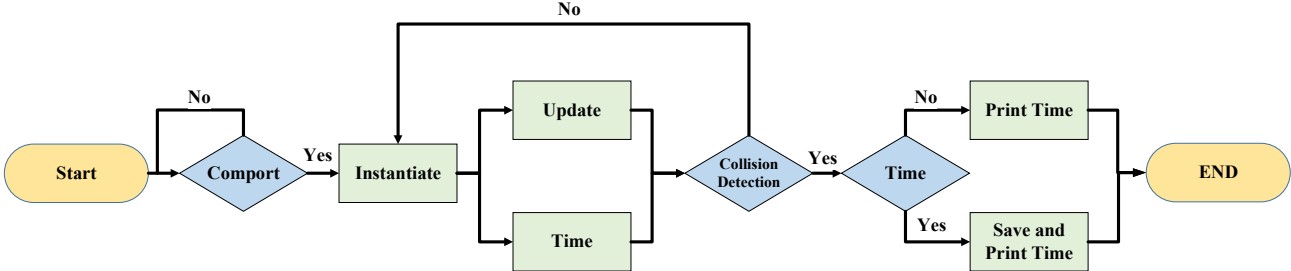

**Figure 17.** VR system flowchart.

### 4.3. The App

The proposed system employs MIT App Inventor to design a mobile app for convenient observation and recording of the user. The app communicates with the sensing end via Bluetooth, and is workable without being connected to a VR headset. The start menu displays the connection, settings, and related app information (see Figure 18A). The user's name, gender, and age are shown on the connection page. The user could choose to connect the smartphone to either both hands or one hand. When the corresponding sensing end is connected, the current EMG diagram is displayed instantly, so that the users can check their muscle conditions. Figure 18B shows the app's state after the right-hand sensing end is connected. The data could be saved during the connection, and the EMG signals could be automatically saved in the micro database by clicking the save button. On the other hand, data saving could be stopped immediately by clicking the stop button or the "Connection Off" button. The previously-accessed EMG data could be queried on the "View Data" page. Figure 18C shows the EMG data found using "View Data". The user information could be modified in the setting interface (Figure 18D). In addition to EMG signals, the app could collect biomedical data (ECG, EEG, and blood oxygen), making it highly suitable for long-term collection and observation.

On the menu displayed when the users open the app, the main operation includes the "Connect" and "Set" options. "View Data" or "Scan" could be selected in the connect setting. "View Data" is for EMG signal record queries, by which the saved EMG signal data could be observed. "Scan" is to view the list of nearby Bluetooth devices, through which the wearable system with intelligent and wireless features could be selected and the corresponding Bluetooth connection could be established. The current physiological condition could be observed after the connection is verified, the data could be saved by clicking "Save" while in the connected state, and the Bluetooth connection could be disconnected by clicking "Disconnect". The user data could be input in the Set part, and "Enter" could then be chosen to complete the setting, or the user could choose "Clear" to clear the user data.

The operation response time of this system is shown in Table 4. There are 320 pixels in one EMG image and it takes 0.1 s for one pixel, so totaling 30 s to draw the whole EMG image and the sample rate of the hardware capturing EMG signals is 300 Hz.

**Table 4.** The system operation response time.

| Event | Time |
| --- | --- |
| EMG image update | 30 (s) |
| Hardware sampling | 3.3 (ms) |

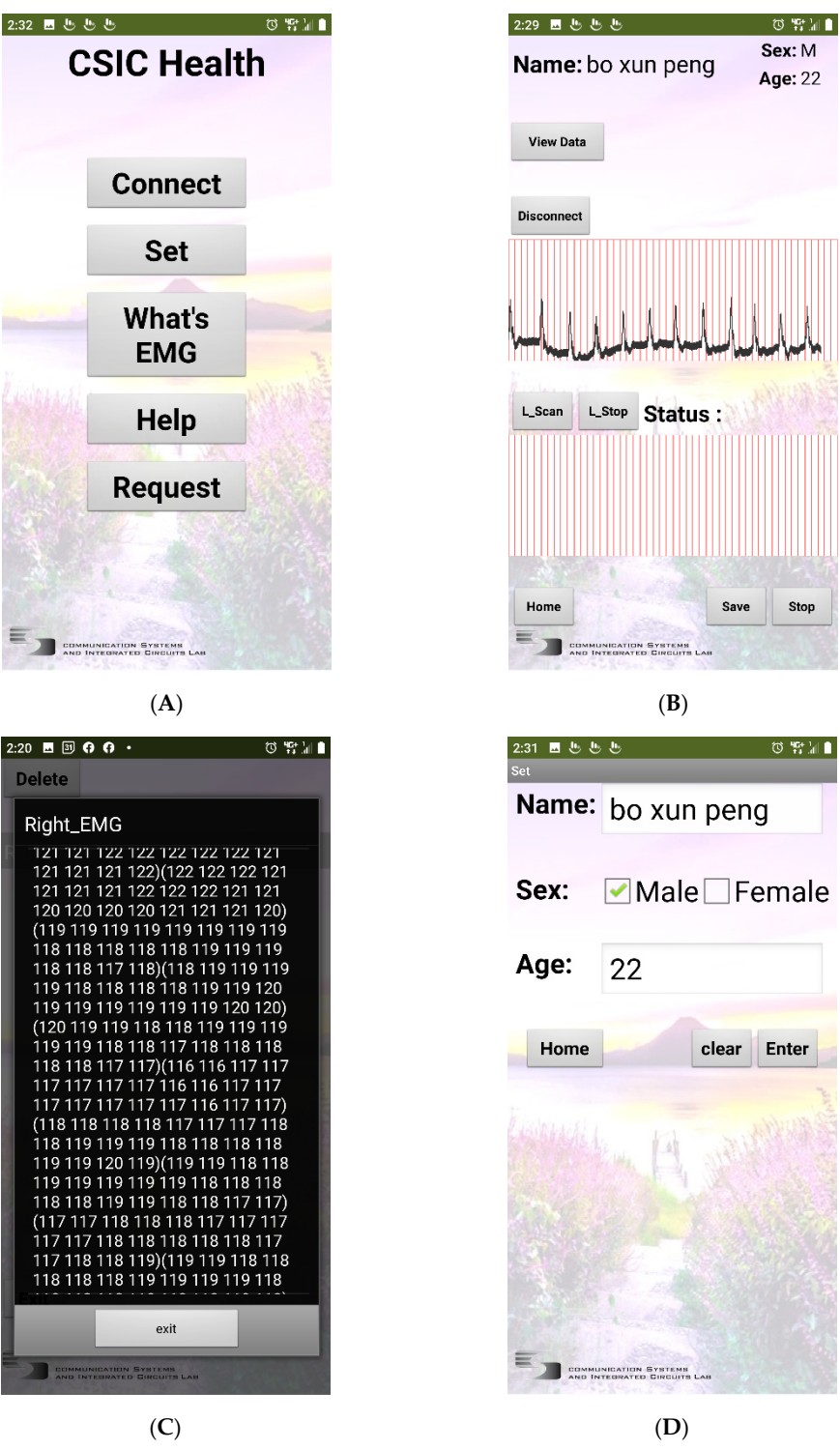

**Figure 18.** Mobile app usage pictures. (**A**) Home page, (**B**) Connection interface, (**C**) Reading the EMG data, (**D**) User data settings.

## 5. Conclusions

This study developed an intelligent wireless wearable system in combination with IoT and VR interactive games in order to allow users to monitor their physiological conditions anytime through wearable and IoT devices. This system combined EMG with VR, so as to enhance users' enjoyment during exercise and rehabilitation. The EMG peak detection accuracy of the ASM S & C algorithm proposed in this paper was higher than that of the S & C and ACD algorithms by 20.98% and 23.71%, respectively. A mobile app was also

designed for long-term physiological signal collection and observation. In our future study, more biomedical sensors will be integrated, and a fatigue test and ECG diagnosis functions will be added so that users' physiological conditions can be better visualized to achieve health care purposes.

**Author Contributions:** Conceptualization, K.-H.L. and B.-X.P.; methodology, K.-H.L.; software, K.-H.L. and B.-X.P.; validation, K.-H.L. and B.-X.P.; formal analysis, K.-H.L.; investigation, K.-H.L. and B.-X.P.; data curation, B.-X.P.; writing—original draft preparation, K.-H.L. and B.-X.P.; writing—review and editing, K.-H.L.; visualization, K.-H.L. and B.-X.P.; supervision, K.-H.L.; project administration, K.-H.L.; funding acquisition, K.-H.L. All authors have read and agreed to the published version of the manuscript.

**Funding:** This research was supported in part by the ministry of science and technology in Taiwan under the grant numbers MOST 109-2637-E-150-002 and MOST 110-2637-E-150-011.

**Informed Consent Statement:** Informed consent was obtained from all subjects involved in the study.

**Data Availability Statement:** The datasets generated during and/or analyzed during the current study are available from the corresponding author on reasonable request.

**Conflicts of Interest:** The authors declare no conflict of interest.

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
