# Peer review of "Wearable Technology and Visual Reality Application for Healthcare Systems"

_electronics, doi:10.3390/electronics11020178_

Round 1

Reviewer 1 Report

The paper “Wearable Technology and Visual Reality Application for Healthcare Systems” by Kuang-Hao Lin and Bo-Xun Peng, describes a portable system able to measure EMG signals to be processed by a mobile app.

The paper is written in a clear form but several parts need to be explained in a more detail in order to make the work more comprehensible to a wider readership. Below I suggest some improvements that I think are needed before considering the article for publication.

Flow diagrams on fig. 8 and fig. 9 seem to refer to a hardware solution. Is it true? How did the authors implemente the represented block diagrams? If not (as I think) I'm a little confused by their representation if the diagrams refer to code that will be executed by a microprocessor.

In addition, the flow diagram reported in fig. 10 must be described before the code reported in section 3.1.

Equation (2) makes no sense. Please, describe the algorithm used for sorting or remove the equation: it does not add any information.

Row 115: why the authors collected 104 samples? Please explain.

Row 121: why the authors removed 3 samples? Please explain.

Fig. 4: is the scale factor D? If it is so, please report D on x axis. How did the authors calculate the accuracy? Please, report the formula when you summarize the results in the graph. In addition, why did the authors display the accuracy for a scale factor in the 0.3 – 0.9 range? How did they choose this range?

Equation (9) must be reported in a mathematical form. It is identical to what was reported for the code.

Row 226: how did the authors calculate the lambda value?

From row 283 to row 294, the authors repeat the concept expressed above. Please reword the sentences.

Row 296: what do the author mean by mentioning that “EMG graphs were simulated by MATLAB”? What is the meaning of “simulated”?

Did the authors develop the app briefly described in section 4.3? If it is so, do the authors make it available?

The flowchart shown in Fig. 18 is not clear. It does not execute a loop. So how does the application end? More importantly. The flowchart shown does not introduce any useful information and it can be removed.

Finally, in the text please use italics to identify the variables used in the code.

Several typos must be corrected (see, for example, eq. 5 and eq. 6).

Author Response

  • Flow diagrams on fig. 8 and fig. 9 seem to refer to a hardware solution. Is it true? How did the authors implemente the represented block diagrams? If not (as I think) I'm a little confused by their representation if the diagrams refer to code that will be executed by a microprocessor.

Reply: Thank you for your comments. MCU is used in this manuscript for development and design. Figure 8 and Figure 9 have been replaced by the MCU processing flowcharts.

  • In addition, the flow diagram reported in fig. 10 must be described before the code reported in section 3.1.

Reply: The system flow diagram has been placed before the code reported.

  • Equation (2) makes no sense. Please, describe the algorithm used for sorting or remove the equation: it does not add any information.

Reply: Equation (2) has been removed. It has been stated in ACD that general sorting was used for a greater value in comparison.

  • Row 115: why the authors collected 104 samples? Please explain.

Reply: In order to capture enough EMG signals for detection, this system captured 10,000 samples. The “104” in the manuscript is a clerical error, and should be 10^4. The correction has been made in the manuscript.

  • Row 121: why the authors removed 3 samples? Please explain.

Reply: There were signal bounces occasionally in EMG signal detection. In order to eliminate the impacts of high signal bounces, the first three pieces of data in the eight pieces were removed, thereby preventing the values of high signal bounces from affecting the average value.

  • Fig. 4: is the scale factor D? If it is so, please report D on x axis. How did the authors calculate the accuracy? Please, report the formula when you summarize the results in the graph. In addition, why did the authors display the accuracy for a scale factor in the 0.3 – 0.9 range? How did they choose this range?

Reply: Figure 4 is scale factor D, which uses Eq. (7) for the calculation of accuracy. It has been described in summarize the results in the graph. A full analysis on the scale factor D in the range from 0.1 to 1 has been conducted, and the highest accuracy in the scale factor D has been used in the analysis result as the system parameter, hence the selection of 50%.

  • Equation (9) must be reported in a mathematical form. It is identical to what was reported for the code.

Reply: Eq. (9) has been modified into a mathematical form and renamed as Eq. (8).

  • Row 226: how did the authors calculate the lambda value?

Reply: The lambda value in the initial_value calculation is the gathered statistics parameter.

  • From row 283 to row 294, the authors repeat the concept expressed above. Please reword the sentences.

Reply: Sentences in this paragraph have been rewritten, and the sentences of the same meaning have been removed.

  • Row 296: what do the author mean by mentioning that “EMG graphs were simulated by MATLAB”? What is the meaning of “simulated”?

Reply: MATLAB was used to display the result after EMG signals were captured. This sentence was simplified to avoid misunderstanding of readers.

  • Did the authors develop the app briefly described in section 4.3? If it is so, do the authors make it available?

Reply: The app of this system is still under development, and the cooperation with a manufacturer is underway. As a result, the app is not available at this point.

  • The flowchart shown in Fig. 18 is not clear. It does not execute a loop. So how does the application end? More importantly. The flowchart shown does not introduce any useful information and it can be removed.

Reply: Thank you for your comment. The flowchart has been deleted and modification has been made in the description of the manuscript.

  • Finally, in the text please use italics to identify the variables used in the code. Several typos must be corrected (see, for example, Eq. 5 and Eq. 6).

Reply: Thank you for your comment. The corresponding typos have been corrected.

Reviewer 2 Report

This manuscript proposed an average slope maximum based approach to detect EMG signal peaks. There are several major issues in this manuscript that require major re-write before further consideration.

  1. The novelty of this manuscript needs further explanation.
  2. More recent related works should be discussed (the S & C algorithm was developed more than 20 years ago).
  3. More details of the improvements to the S & C algorithm should be provided.
  4. More state-of-the-art algorithms should be compared to validate the effectiveness of the proposed algorithm.

  5. The developed VR game is uncorrelated to the main research goal of this manuscript.

Author Response

  • The novelty of this manuscript needs further explanation.

Reply: The novelty of this manuscript in “introduction” has been modified. The realization architecture of low power consumption and low cost provided in this study mainly targets the irregularity at EMG peaks to improve the operational accuracy and smoothness of VR games.

This study proposed the average slope maximum So and Chan (ASM S & C) algorithm. By improving the capture requirements of So and Chan’s (S & C) maximum slope, the EMG peak signal detection was enhanced greatly. The operation sensitivity could be enhanced when the EMG signal was combined with a VR app.

  • More recent related works should be discussed (the S & C algorithm was developed more than 20 years ago).

Reply: AI has been widely is most often used in the identification and analysis of physiological signals in recent years [14,15], but it is complicated. Therefore, this study developed an innovative realization architecture with low power consumption and low cost. The proposed system specifically tarts the irregularities of EMG peaks to improve the operational accuracy and smoothness of VR games.

  • More details of the improvements to the S & C algorithm should be provided.

Reply: Thank you for your comment. The initial_slope_maxi in the S & C algorithm is captured in a preset period of cycle time. In ECG with a cycle that changes slightly, accurate initial_slope_maxi can be obtained through S & C after adjusting the cycle based on the ECG of different persons. However, for EMG with irregular changes, detection based on a fixed cycle is not feasible. In order to solve this problem, the capture requirements of the maximum slope were changed in this study, as shown in algorithm 2: ASM S & C algorithm. The features and performance of ASM S & C algorithm have also been stated in the manuscript.

  • More state-of-the-art algorithms should be compared to validate the effectiveness of the proposed algorithm.

Reply: While AI has been widely used in recent physiological identification and analysis, it is very complicated and costly for calculation. For the purpose of popularizing the design and application, this study omitted the introduction of advanced technologies in the analysis process.

  • The developed VR game is uncorrelated to the main research goal of this manuscript.

Reply: This manuscript developed a VR interactive game with smart wireless wearable technology for healthcare of elderly users. The proposed wearable system uses its intelligent and wireless features to collect EMG signals in a VR app. In order to improve the operational accuracy and smoothness of VR games, ASM S & C algorithm was proposed in this manuscript to increase the accuracy of EMG peak detection up to 95%.

Reviewer 3 Report

This article describes a virtual reality (VR) interactive game for healthcare systems that incorporates smart wireless wearable IoT technologies. The authors' method to merging electrocardiography (ECG) with neuromorphic computing in an Internet of Things edge device is both creative and powerful. The coupling with VR, in particular, is novel and demonstrates its intriguing application potential in the healthcare system. The article is well-written, with a clear logic, a well-structured framework, and extensive data. Meanwhile, this text is in desperate need of revision. For instance, some sentences in this article are unclear, and several illustrations need more explanation. Acceptance is suggested after a minor revision.

1) In a real-time system, response time is a critical characteristic. How long does it take for the system as a whole to process an action and finish the EMG graphs? How much time does hardware consume? And how much time need to process signal on software (MIT App)?

2) Introduction of this manuscript very short and limited to only AI enabled biomedical signals recognition system. In recent years, artificial intelligence (AI) or neuromorphic headwear-based recognition systems have advanced rapidly (Nat Commun 12, 5378 (2021); Small 2021, 17, 2103543; Nature 575, 473–479 (2019); Adv. Electron. Mater. 2021, 7, 2100142). It is strongly advised that authors should discuss the recent advances in neuromorphic headwear-based recognition systems based on above mentioned reference.

3) When developing artificial intelligence, feature engineering is an essential component. Please elaborate on some of the most relevant elements of this study that were gleaned from the raw data. As well as the mathematical techniques used to achieve these traits.

4) In terms of biological signal recognition, there are a number of AI systems that are comparable. There should be some kind of comparison between this work and other works in the form of a table in order to highlight its important benefits.

5) There seems to be a discrepancy between the peak pulse and the actual EMG signal output (delayed response). Please explain why this is happening.

Author Response

  • In a real-time system, response time is a critical characteristic. How long does it take for the system as a whole to process an action and finish the EMG graphs? How much time does hardware consume? And how much time need to process signal on software (MIT App)?

Reply: Thank you for your comment. The response time has been stated in Table 3 of Chapter 4.

  • Introduction of this manuscript very short and limited to only AI enabled biomedical signals recognition system. In recent years, artificial intelligence (AI) or neuromorphic headwear-based recognition systems have advanced rapidly (Nat Commun 12, 5378 (2021); Small 2021, 17, 2103543; Nature 575, 473–479 (2019); Adv. Electron. Mater. 2021, 7, 2100142). It is strongly advised that authors should discuss the recent advances in neuromorphic headwear-based recognition systems based on above mentioned reference.

Reply: Thank you for your comment. The discussion on relevant references has been added.

  • When developing artificial intelligence, feature engineering is an essential component. Please elaborate on some of the most relevant elements of this study that were gleaned from the raw data. As well as the mathematical techniques used to achieve these traits.

Reply: In this study, the physiological signal capture module was used to capture EMG voltage signals, and the ASM S & C algorithm was used to detect EMG peak signals, which realized low-cost hardware architecture.

  • In terms of biological signal recognition, there are a number of AI systems that are comparable. There should be some kind of comparison between this work and other works in the form of a table in order to highlight its important benefits.

Reply: AI has been widely used in the majority of recent physiological identification and analysis, however, it is complicated and costly for calculation. For the purpose of popularizing the design and application, this study did not introduce advanced technologies in the analysis process.

  • There seems to be a discrepancy between the peak pulse and the actual EMG signal output (delayed response). Please explain why this is happening.

Reply: ECG signal capturing by the sensor module was to capture smooth data, and the data would go through a moving average circuit, resulting in delayed response, which, however, does not affect the judgment of EMG peak.

Reviewer 4 Report

Authors proposed EMG-based IoT systems using average slope maximum So and Chan algorithm. Therefore, it can be connected to Virtual Reality application. English grammar looks fine.  However, literature search is simple to be understood. Spectrum analysis is missing. There are some suggestive comments to be improved before publication. 

  1. Literature search for EMG-based IoT system with VR games need to be provided in the introduction section.
  2. In Figure 12, authors need to provide the spectrum data to show the frequency analysis.
  3. Figure 17 quality is low.
  4. Please use abbreviated journal names in the reference section.
  5. In Figure 12, authors had better show some variance in the measured data to show accuracy.
  6. In Figure 7, Voltage (V) could be Voltage (Vp-p).
  7. In Algorithm 2, is there any ranges for n ?
  8. Authors does not show how to construct the EMG system in detail from Figure 11. 

Author Response

  • Literature search for EMG-based IoT system with VR games need to be provided in the introduction section.

Reply: Thank you for your comment. References for [13,14] have been added.

  • In Figure 12, authors need to provide the spectrum data to show the frequency analysis.

Reply: Figure 13 is the spectrum analysis of Figure 12. Figure 13-(a) is the spectrum distribution of the EMG signals captured. Figure 13-(b) is the spectrum distribution of the EMG signals after moving average. As seen, there are obviously fewer noises in the spectrum distribution after moving average.

  • Figure 17 quality is low.

Reply: Thank you for your comment. Figure 17 has been replaced with a high-resolution image.

  • Please use abbreviated journal names in the reference section.

Reply: Thank you for your comment. It has been corrected according to your comment.

  • In Figure 12, authors had better show some variance in the measured data to show accuracy.

Reply: In Figure 12, EMG signals are obviously cleaner after filtration, improving the ability of ASM S & C to identify them.

  • In Figure 7, Voltage (V) could be Voltage (Vp-p).

Reply: Thank you for pointing this out. It has been corrected according to your comment.

  • In Algorithm 2, is there any ranges for n ?

Reply: The n of Algorithm 2 was 1000 samples, as stated in Line 261.

Each sample was 3.3 msec, and 1,000 pieces of data (about 3.3 sec) were captured for analysis during the initial setting of the initial_value, so as to capture stable initial_value data.

  • Authors does not show how to construct the EMG system in detail from Figure 11. 

Reply: In Figure 11, the TriAnswer platform was strapped to the user’s wrist, and it captured EMG signals via a patch. EMG signals were transmitted to the cloud via the Bluetooth wireless interface and displayed in a smartphone app.

Round 2

Reviewer 1 Report

In the new version, authors upgraded the content of the paper according to given suggestions. The logical order of their work is now more clear. However, some corrections have to be still made.

Flow diagram reported in Fig. 3 is not coherent with the text and it results incorrect.

For moving average, in equation 1 sum needs to be done between P=0 and P=T.

In the new sentences, code variables have to be reported in italics.

In the text, several times authors do not insert a space character between values and units.

Few typos have to be corrected.

Author Response

In the new version, authors upgraded the content of the paper according to given suggestions. The logical order of their work is now more clear. However, some corrections have to be still made.

Reply: Thank you for your comments.

1) Flow diagram reported in Fig. 3 is not coherent with the text and it results incorrect.

Reply: Thank you for your comments. Figure 3 has been redrawn and re-explained in line 90 of the text.

2) For moving average, in equation 1 sum needs to be done between P=0 and P=T.

Reply: Thank you for your correction. Equation 1 has been corrected according to your comment.

3) In the new sentences, code variables have to be reported in italics.

Reply: Thank you for your correction. The equation 3 variables have been corrected to italics.

4) In the text, several times authors do not insert a space character between values and units.

Reply: Thank you for your correction. It has been corrected according to your comment.

Line 105: 500Hz => 500 Hz

Line 187: 500Hz => 500 Hz

Line 316: 60Hz => 60 Hz

Line 399: 300Hz => 300 Hz

5) Few typos have to be corrected.

Reply: Thank you for your comments. Some typos have been corrected.

Such as the second paragraph of Introduction has been corrected to “an epilepsy prediction system”.

In Line 272, the punctuation “the gathered statistics parameter.” has been corrected.

Reviewer 2 Report

The authors did not fully address the comments in the previous round:

  1. The comparison to state-of-the-art methods, which is a necessary step in verifying the effectiveness of the proposed work, is still missing.
  2. The authors claimed that their proposed method has "low power consumption and low cost" compared to the AI-based method, but neither theoretical nor empirical analysis is provided.
  3. The comparison methods do not have to be AI-based approaches, the recent developments based on S & C algorithm also count.
  4. Although some recent papers are listed, they were not discussed in detail and compared.

Author Response

The authors did not fully address the comments in the previous round:

Reply: Thank you for your comments.

1) The comparison to state-of-the-art methods, which is a necessary step in verifying the effectiveness of the proposed work, is still missing.

Reply: Recent common EMG analyses and studies focused on AI applications. For example, Sugiarto [15] analyzed the surface EMG (sEMG) and high-density EMG (HD-EMG) of the neck muscle (designated muscle) and used Convolutional Neural Network (CNN) for training to correct the end-to-end delay of the VR system and alleviate negative effects like motion sickness. Sugiarto [15] also employed three pairs of wireless sEMG sensors from Delsys Trigno (Delsys, MA, USA) as the software. Pancholi [16] analyzed the peak average power (PAP) of EMG and adopted Linear Discriminant Analysis (LDA) and Quadratic Discriminant Analysis (QDA) for training and prosthetics. The analysis of Pancholi [16] has been done using MATLAB 2015a in the i7 core. Raurale [17] employed subsequently classified to classify and identify eight kinds of EMG actions. The system is also shown to operate in real-time on an ARM Cortex A-53 embedded processor suitable for housing in an EMG wearable device. Besides, Wang [18] modified the So and Chan algorithm to increase the accuracy of ECG peak detection to 99.16% from 94.61% and used FPGA for verification.

2) The authors claimed that their proposed method has “low power consumption and low cost” compared to the AI-based method, but neither theoretical nor empirical analysis is provided.

Reply: Thank you for your comments. The description of low power consumption and low cost in the paper was not accurate, and this study mainly provides a less complex EMG peak detection method, so we revised the description in the paper.

3) The comparison methods do not have to be AI-based approaches, the recent developments based on S & C algorithm also count.

Reply: S & C in recent papers is as shown in Table 1, which was mainly used for ECG signal detection, but its accuracy was significantly lower when used for EMG signal detection, as shown in Table 3, so in this paper, we proposed that ASM S & C could improve the S & C already used for ECG and make it applicable for EMG peak detection.

4) Although some recent papers are listed, they were not discussed in detail and compared.

Reply: The comparison of recent papers on EMG signal detection was made in Introduction, and the comparison results were listed in Table 1. In this paper, we used VR for EMG signals to accurately detect EMG peaks without highly complex characteristic identification so that we could realize it only using microcontrollers when using this ASM S & C.

Round 3

Reviewer 2 Report

The authors have addressed my previous comments.